# Association of Radiation Dose to the Amygdala–Orbitofrontal Network with Emotion Recognition Task Performance in Patients with Low-Grade and Benign Brain Tumors

**DOI:** 10.3390/cancers15235544

**Published:** 2023-11-23

**Authors:** Sara J. Hardy, Alan Finkelstein, Michael T. Milano, Giovanni Schifitto, Hongying Sun, Koren Holley, Kenneth Usuki, Miriam T. Weber, Dandan Zheng, Christopher L. Seplaki, Michelle Janelsins

**Affiliations:** 1Department of Radiation Oncology, University of Rochester, Rochester, NY 14620, USA; michael_milano@urmc.rochester.edu (M.T.M.); dandan_zheng@urmc.rochester.edu (D.Z.); michelle_janelsins@urmc.rochester.edu (M.J.); 2Department of Neurology, University of Rochester Medical Center, Rochester, NY 14642, USA; giovanni_schifitto@urmc.rochester.edu; 3Department of Biomedical Engineering, University of Rochester, Rochester, NY 14627, USA; alan_finkelstein@urmc.rochester.edu; 4Center for Advanced Brain Imaging and Neurophysiology, University of Rochester Medical Center, Rochester, NY 14642, USA; 5Department of Imaging Sciences, University of Rochester Medical Center, Rochester, NY 14642, USA; koren_holley@urmc.rochester.edu; 6Department of Surgery, Supportive Care in Cancer, University of Rochester Medical Center, Rochester, NY 14642, USA; hongying_sun@urmc.rochester.edu (H.S.); miriam_weber@urmc.rochester.edu (M.T.W.); 7Department of Obstetrics and Gynecology, University of Rochester Medical Center, Rochester, NY 14642, USA; 8Department of Public Health Sciences, University of Rochester Medical Center, Rochester, NY 14642, USA; christopher_seplaki@urmc.rochester.edu; 9Office for Aging Research and Health Services, University of Rochester Medical Center, Rochester, NY 14642, USA

**Keywords:** amygdala, social cognition, cranial radiation, orbitofrontal cortex, radiation-related cognitive decline, emotion recognition task

## Abstract

**Simple Summary:**

Patients with brain tumors often experience changes in memory and other aspects of thinking. Many may also have difficulty with social cognition, affecting the abilities that facilitate social behavior and maintain social relationships; however, the data are limited. Whether treatments such as cranial radiation can impact social cognition is not well-studied. We sought to understand how radiation dose exposure to the amygdala–orbitofrontal network, which subserves social cognition and emotion recognition, impacted performance on an emotion recognition task. We found that radiation dose to the amygdala and associated structures was associated with performance on an emotion recognition task, including longer response times with increasing radiation doses. Radiation techniques that reduce the dose to the amygdala-orbitofrontal network may decrease side effects for patients receiving cranial radiation.

**Abstract:**

Background: Although data are limited, difficulty in social cognition occurs in up to 83% of patients with brain tumors. It is unknown whether cranial radiation therapy (RT) dose to the amygdala–orbitofrontal network can impact social cognition. Methods: We prospectively enrolled 51 patients with low-grade and benign brain tumors planned for cranial RT. We assessed longitudinal changes on an emotion recognition task (ERT) that measures the ability to recognize emotional states by displaying faces expressing six basic emotions and their association with the RT dose to the amygdala–orbitofrontal network. ERT outcomes included the median time to choose a response (ERTOMDRT) or correct response (ERTOMDCRT) and total correct responses (ERTHH). Results: The RT dose to the amygdala–orbitofrontal network was significantly associated with longer median response times on the ERT. Increases in median response times occurred at lower doses than decreases in total correct responses. The medial orbitofrontal cortex was the most important variable on regression trees predicting change in the ERTOMDCRT. Discussion: This is, to our knowledge, the first study to show that off-target RT dose to the amygdala–orbitofrontal network is associated with performance on a social cognition task, a facet of cognition that has previously not been mechanistically studied after cranial RT.

## 1. Introduction

Cognitive dysfunction occurs in up to 91% of patients with brain tumors [1,2,3,4,5,6,7] and has a major impact on quality of life. Even mild cognitive dysfunction can affect daily activities, the ability to work, and social roles [8]. Higher scores on cognitive assessment batteries are positively associated with quality of life and patient well-being [9].

Baseline cognitive deficits are common in patients with brain tumors; however, cancer-directed therapies, including radiotherapy (RT), can further disrupt cognition. Cognitive sequelae from cranial RT, often termed RT-induced cognitive decline (RICD), has recently received more attention. RICD is observed in more than 30% of patients at 4 months after partial or whole brain RT and in more than 50% at 6 months [10]. RICD is particularly important in patients with low-grade or benign tumors who are expected to have long-term survival. Considerable efforts have been directed toward understanding and preventing RT-induced cognitive decline (RICD), a serious late effect of RT [11,12,13]. To date, multiple mechanisms underlying RICD have been elucidated, including damage to sites of neurogenesis [14,15], neuroinflammation [16,17], neuronal dysfunction [18], and vascular changes [19,20,21].

However, social cognition, which underlies the abilities that facilitate social behavior and maintenance of social relationships, is not frequently tested or addressed in patients with brain tumors. Social cognition includes processing, memorizing, analyzing, and applying information about other people and social situations [22]. It describes the skills used to understand what others think and intend [23]. While many skills are components of social cognition, one key piece is recognizing human emotional states based on facial or vocal cues. Recognizing human emotions is a fundamental skill and a building block for more complex skills, such as making socially appropriate decisions [24].

The ability to recognize emotional states can be tested using an emotion recognition task (ERT), which involves the presentation of faces with six basic facial expressions, including anger, disgust, fear, happiness, sadness, and surprise [25]. Cambridge Cognition provides a computerized version of this test as part of its neurocognitive battery [26]. Though it has received limited study, Goebel et al. reported that there was social cognition impairment in 83% of patients with brain tumors [9], indicating that this is a critical domain to measure and understand in patients with brain tumors.

Social cognition involves diverse skills that rely on multiple brain networks. However, many social cognition functions are mediated through anatomic and functional connections to the amygdala, a paired limbic structure located in the temporal lobe with extensive connections to various cortical and subcortical regions. The amygdala is implicated in a wide variety of behavioral processes, including fear-related processes [27,28,29], social perception, learning, decision-making, social memory, and interaction [30]. Patients with bilateral amygdala lesions show impairment in recognizing and responding to social stimuli [30]. The amygdala, in combination with the orbitofrontal cortex, also subserves emotion recognition [30]. Overall, available data suggest that the amygdala is essential for many aspects of social cognition.

In recent years, technology has evolved, and the RT dose can be refined more precisely; concurrently, evidence-based data have been developed to minimize doses to normal structures (susceptible to RT-mediated toxicity) without compromising treatment efficacy. For example, intensity-modulated RT (IMRT) and image-guided RT (IGRT), used to minimize doses to normal intracranial structures (e.g., optic chiasm, cochlea, and brainstem), are now the standard of care for primary brain tumor treatment. These treatment modalities allow radiation oncologists to create plans that keep important intracranial structures below doses that are likely to cause toxicity. However, knowledge of the probability of a specific toxicity at an RT dosage is needed to integrate the parameter into RT planning. While this requires generating normal tissue complication probability models, the first step is understanding whether RT dose to a structure is associated with toxicity.

Significant interest has arisen in applying the idea of normal tissue sparing to alleviate issues such as cognitive impairment; however, the implementation of this concept remains preliminary. In a recent phase III trial, NRG Oncology CC001 showed that reducing the RT dose to the bilateral hippocampi (important structures in learning and memory) during whole brain RT reduced the risk of cognitive decline at 6 months from 68.2% to 59.5% [31]. However, little is known about the cognitive consequences of RT doses on other structures in the brain. Currently, the hippocampus is the only intracranial structure for which validated dose constraints to prevent cognitive decline are used in standard RT treatment planning, and further research is needed to better understand the impact of radiation on cognitive processes subserved by other anatomical structures in the brain [32,33,34,35].

During RT, for brain tumors, there can be unintended or off-target doses to the amygdala and networked structures such as the medial and lateral orbitofrontal cortex. The mean RT dose to the amygdala correlates with amygdala volume loss on MRI [36]. A recent study also showed that, after RT, atrophy in the amygdala was associated with poorer visuospatial memory and emotional well-being [37]. However, more research is needed to understand the consequences of off-target RT dose to the amygdala and associated networks. The amygdala plays a central role in social cognition, including emotional learning, emotion recognition, and regulation, and there may be functional implications where RT dose to the amygdala adversely impacts emotional management.

Our specific aim was to evaluate whether RT dose to the amygdala and other networked structures impacted social cognition, as measured by performance on an ERT. We evaluated three outcomes: the median response time for correct responses (ERT overall correct median reaction time or ERTOMDCRT), the median response time for all responses (ERT overall median reaction time or ERTOMDRT), and the total correct responses (ERT total hits or ERTTH). It is common in the literature for there to be no differences in total correct responses but changes in median reaction time [38,39]; therefore, as our primary outcome, we chose to focus on the change in the ERTOMDCRT with ERTOMDRT and ERTTH as secondary outcomes.

## 2. Materials and Methods

### 2.1. Patients and Procedures

Fifty-one patients with benign or low-grade brain tumors planned to receive partial brain RT, either hypofractionated (>2 Gy/fraction) or conventionally fractionated (1.8–2 Gy/fraction), were recruited at the Wilmot Cancer Institute. The study protocol has previously been published in a separate paper [40], and a manuscript with all primary cognitive outcomes is planned with additional follow-up data from participants in the cohort. Participants underwent evaluations at baseline, 6-month, and 12-month time points, including an ERT through Cambridge Cognition. For this secondary analysis, we focused on 38 patients with baseline and 6-month outcomes on the ERT (Figure 1). 

### 2.2. Measures

The calculation of the RT dose map, delineation of intracranial structures, and calculation of the mean RT dose to intracranial structures, as well as RT plans for radiosurgery (hypofractionated RT plans delivered in 5 or fewer fractions), were created using Brainlab Elements^®^ planning software 3.0 (Brainlab, Munich, Germany). Plans for conventionally fractionated RT were created using Varian Eclipse^®^ treatment planning software 15.6 (Varian Medical Systems, Palo Alto, CA, USA). For consistency, all RT dose maps were recalculated in Eclipse for all patients using a 1 mm x 1 mm grid. Each subject had completed a high-resolution, T1-weighted (T1w) MRI brain scan before RT on a 3T GE Discovery 750 MRI system (Milwaukee, WI, USA) equipped with an 8-channel head coil. T1w images were acquired using a 3D BRAVO FSPGR sequence with the following parameters: repetition time (TR) = 8.2 ms, echo time (TE) = 3.2 ms, field of view (FOV) 256 mm^2^, resolution 1 × 1 × 1 mm^3^. All image processing was completed within URMC servers in the Center for Integrated Research Computing (CIRC) using BHWARD, a HIPAA-compliant server. T1w images were processed by first masking out the tumor using the gross target volume (GTV) as contoured by the primary radiation oncologist from the RT structure set [41], delineated on planar MRI and CT imaging, using nibabel (https://nipy.org, version 3.1.1). This was achieved by mapping the GTV on CT images and performing an affine transform to register the T1w images to the patient-specific CT images. Thereafter, segmentation was performed using the tumor-masked T1w image in Freesurfer (version 6.0.0, http://surfer.nmr.harvard.edu). Briefly, processing included skull-stripping and the removal of non-brain tissue, motion correction, intensity normalization, automated Talairach transformation, white matter segmentation, and cortical parcellation using the Desikan–Killiany atlas [42], which includes cortical and subcortical regions of interest (ROIs) including the amygdala, medial orbitofrontal cortex, and lateral orbitofrontal cortex [43]. The RT dose map was first scaled and mapped with CT images using pydicom (version 1.4). T1w images were registered to a patient-specific CT space using FMRIB’s Linear Image Registration Tool (FLIRT) [44]. Patient-specific parcellations derived from the Desikan–Killiany [42] atlas using Freesurfer were registered with the RT dose map. The mean RT doses were extracted from each ROI, and the 2 Gy/fraction equivalent dose (EQD2) was calculated using the linear quadratic model [45], with an α/β equal to 2 Gy, to model the radiosensitivity of normal brain tissue [46]. 

#### 2.2.1. ERT Outcomes

The Cambridge Cognition tests included the ERT, which consists of a series of morphed faces depicting a continuum of expression magnitude for six basic emotions: happiness, surprise, fear, disgust, anger, and sadness. One hundred-and-eighty faces are displayed for 200 ms each, and after each display, participants are required to choose the correct emotion, as fast as they can, among the six basic emotions displayed. Measures of emotion discrimination performance and reaction time (RT) or time to choose an emotion are quantified. This study examined the ERTOMDCRT, the overall median time in milliseconds (ms) to select a correct response (the primary outcome), as well as the ERTOMDRT, the median time to select a response (correct or incorrect), and the ERTTH, the total correct responses (both secondary outcomes). Increases in the ERTOMDCRT and ERTOMDRT and decreases in the ERTTH were consistent with worse performance. Neuropsychological testing was administered by trained study coordinators using a standardized testing manual; study coordinators were supervised by members of the study team with expertise in neurology, neuropsychology, and cognitive science. All testing was performed in a quiet, comfortable room without distractions.

#### 2.2.2. Patient Characteristics

Clinical variables were recorded, including age, education level, comorbidities including diabetes, hypertension, autoimmune disease, tumor hemisphere, tumor site, tumor pathology, prior surgeries, employment status, smoking status, alcohol use, sex/gender, hypopituitarism, menopausal status, steroid use, the use of medications that can affect cognition and mood, and exposure to chemotherapy. For this study, we examined age, depression (evaluated using PHQ2), and sex as covariates since these variables have been shown to affect performance on ERT [47,48], as well as brain tumor-specific variables that may impact cognitive testing, including tumor histology and laterality [49].

### 2.3. Statistical Analyses

We analyzed 38 patients with complete data for baseline and 6-month timepoints. We conducted descriptive analyses for all variables and distributions. We evaluated associations between radiation doses to the right and left amygdala, lateral orbitofrontal cortex, and medial orbitofrontal cortex and ERTOMDCRT (primary outcome), as well as ERTOMDRT and ERTHH (secondary outcomes) at 6 months. We anticipated that there may be practice effects leading to an improvement in baseline scores between the baseline and 6-month timepoints; however, this is true for both patients who receive high and low RT doses to the amygdala; therefore, a comparison can be made between these two groups without accounting for this statistically. In this study, we examined the mean RT dose to the amygdala–orbitofrontal network, calculated by averaging doses across all of the individual voxels of the contoured structure. The amygdala, lateral orbitofrontal cortex, and medial orbitofrontal cortex are paired structures. Thus, we reviewed the mean RT dose to both right and left structures for each participant and recorded the highest value as the “highest mean dose.” For each outcome (change in ERTOMDCRT, ERTOMDRT, and ERTHH), we additionally generated regression trees using proc hpsplit (RSS criterion, cost-complexity pruning, and 10-fold cross-validation) to examine the optimal split for the highest mean RT dose to the paired structure that minimizes the residual sum of squares. Then, we compared the mean longitudinal change from baseline to 6 months in ERTOMDCRT, ERTOMDRT, and ERTHH for these dichotomous categories. The change scores were tested for normality using the Shapiro–Wilk, Kolmogorov–Smirnov, Cramér–von Mises, and Anderson–Darling tests, and parametric and non-parametric tests were used to compare means between the RT dose groups.

We then created a linear regression using PROC REG in SAS 9.4 to examine the association between the highest mean RT dose to the amygdala (x) and the 6-month ERTOMDCRT (y). Residuals were visually inspected, and White’s test was performed to test for heteroskedasticity. To account for potential confounding, we adjusted for age, depression, and sex, which have been shown to affect performance on ERT [47,48], as well as tumor-specific variables that impact cognitive testing and RT treatment factors, including tumor histology [49], as well as tumor laterality, since patients with right-sided tumors could have more baseline effects on emotion recognition [50].

Doses to the amygdala, lateral orbitofrontal cortex, and medial orbitofrontal cortex were highly correlated and could not be examined together in a standard linear regression. For that reason, a random forest regression tree was created using PROC HPSPLIT in SAS 9.4 to model ERTOMDCRT change from baseline to 6 months as a function of tumor histology, tumor laterality, sex, and dosimetric variables, including the highest mean dose to the amygdala, lateral orbitofrontal cortex, and medial orbitofrontal cortex. A regression tree was grown using RSS criterion, cost-complexity pruning, 10-fold cross-validation, and a minimum leaf size of 8. The regression tree and variable importance were then evaluated.

## 3. Results

Fifty-one subjects completed the baseline assessment, and thirty-eight subjects had 6-month outcome data, resulting in an analysis sample of n = 38. Baseline patient characteristics for the whole cohort and evaluable patients are summarized in Table 1. Of the evaluable patients, 24 (63%) were female, and the remainder were male. The majority of the patients had either cranial nerve schwannomas or meningiomas. Most patients had presumed grade 1 tumors based on imaging only (63%); eight (21%) had pathologically confirmed grade 2 tumors, five (13%) had pathologically confirmed grade 1 tumors, and one patient’s tumor did not have a pathologically assigned grade based on standards for the tumor type. Forty-two percent had left-sided tumors, thirty-two percent had right-sided tumors, and the remainder had midline or bilateral tumors. The median age was 58, and the median score on the PHQ2 was 1 (≥3 is consistent with depression). The majority (84%) were white. Only a small percentage of patients received chemotherapy (8%). The median prescribed RT dose was 23 Gy. Seventeen (45%) of the patients received conventionally fractionated RT, and the remainder received hypofractionated RT. Smoking status, work status, and education level for the cohort are summarized in Appendix A. 

For univariate associations with the primary analysis outcome (6-month ERTOMDCRT), total RT dose, number of RT fractions, tumor histology, age, and baseline scores on ERTOMDCRT were found to be significant (Table 1).

The association between 6-month scores on ERT outcomes and RT dose to the amygdala, lateral orbitofrontal cortex, and medial orbitofrontal cortex was explored using linear regression after adjusting for the baseline score. There was a significant association between worse performance on ERTOMDRT and higher doses to the left amygdala, right and left lateral orbitofrontal cortex, and right and left medial orbitofrontal cortex (Table 2). For the ERTOMDCRT, there was a significant association between greater highest mean RT dose to the amygdala and worse performance on the ERTOMDRT at 6 months (Table 2). For ERTTH, only the mean RT dose to the left medial orbitofrontal cortex was significant (Table 2).

For the whole cohort, ERTOMDCRT and ERTOMDRT decreased and ERTTH increased from baseline to 6 months (Table 3). Outcomes were compared for the dose categories created using regression trees. The six-month change in the ERTOMDCRT was significantly associated with highest mean RT dose to the amygdala ≥ 6.5 Gy and medial orbitofrontal cortex ≥ 8.2 Gy. The six-month change in the ERTOMDRT was significantly associated with highest mean RT dose to the amygdala ≥ 0.6 Gy and to the medial and lateral orbitofrontal cortex ≥ 1.1 Gy (Table 3). Six-month change in ERTTH was significantly associated with highest mean RT dose to the medial orbitofrontal cortex ≥ 17.4 Gy and to the lateral orbitofrontal cortex ≥ 29.7 Gy (Table 3). Change from baseline to 6 months for two representative patients is shown in Appendix A.

The results of the multivariate linear regression and regression trees further examining outcomes are shown in Table 4 and Table 5 and Figure 2. In the multivariate linear regression, baseline ERTOMDCRT and highest mean RT dose to the amygdala were significant. (Table 4). Using a regression tree, variable importance was ranked for importance in predicting changes in ERTOMDCRT scores. The most important variable was the highest mean dose to the medial orbitofrontal cortex; however, tumor histology and laterality were also important (Table 5). The regression tree is shown in Figure 2.

## 4. Discussion

We sought to estimate the association of off-target RT doses to the amygdala–orbitofrontal network with scores on a social cognition task. We hypothesized that doses to the amygdala and networked structures would impact performance on a social cognition task subserved by the amygdala and orbitofrontal cortex. Specifically, we evaluated the association between RT dose to the amygdala–orbitofrontal network and the outcomes of the ERT.

There was a decrease in ERTOMDCRT and ERTOMDRT scores and increase in ERTTH scores for the whole cohort, showing an overall improvement from baseline to 6 months [51]. This is consistent with the practice effect of repeating the task, as seen in the literature. However, patients who received higher doses to the amygdala–orbitofrontal network had worse median response times (Table 3). ERTOMDRT and ERTOMDCRT were impacted at lower doses than ERTTH. Other studies have shown no differences in total hits (ERTTH) due to ceiling effects [38,39]; however, for this cohort, it is also possible that there was less impact on ERTTH since the dose required to cause a decline in that outcome was higher.

Our primary outcome was the ERTOMDCRT, and we performed further analyses to understand its relationship with dosimetric and clinical variables. A linear regression model (Table 3) showed that, on average, patients who received a greater highest mean RT dose to the amygdala took longer to identify the correct emotion on a recognition task when adjusting for age, histology, baseline depression, tumor laterality, and sex. Our study did not compare the impact of higher doses to one vs both amygdalae; due to the high correlation between the right and left amygdala, they could not be examined in a single model in this study. However, since the amygdala is a paired structure, a higher dose to both amygdalae may be associated with worse outcomes than one amygdala. In the literature, higher bilateral hippocampal RT doses result in more decline in verbal learning and memory [52,53]. Moreover, patients with a bilateral amygdala injury can develop Klüver-Bucy, a clinical syndrome characterized by behavioral changes and memory deficits, which does not occur with unilateral amygdala injury [54]. Tumor histology was shown to impact outcomes on the ERT in the univariate analysis but not in the multivariate linear regression (Table 3). While it has not been shown for social cognition tasks specifically, this is consistent with the literature, namely, that the degree of normal brain invasion and tumor growth velocity impact cognitive deficits in patients with brain tumors [55]. Baseline ERTOMDCRT scores were significantly associated with scores at 6 months in both univariate and multivariate analyses, which is consistent with expectations for a repeated cognitive task.

We also created a regression tree to evaluate the relative importance of the dosimetric and clinical variables in a single model for changes in ERTOMDCRT scores. In this model, the RT dose to the medial orbitofrontal cortex was the most important and only dosimetric factor in the final tree. This suggests that sparing structures such as the medial orbitofrontal cortex in addition to the amygdala may be important in RT planning. It also suggests that examining all the structures in a network may allow a more complete understanding of RICD.

Data on social cognition in patients with brain tumors, particularly whether there are changes after treatment, are limited [8]. One prior small study evaluated changes in emotion recognition after whole brain RT [56], and studies have shown performance decreases in patients with glioblastoma [57] and varied brain tumors [58] after surgery. There is a critical need to mechanistically study the impact of cranial RT on social cognition. To our knowledge, this is the first study to show that RT dose exposure to the amygdala and amygdala–orbitofrontal network impacts performance on social cognition tasks. This work complements the recent publication showing that RT doses to the amygdala impact well-being scores as well as memory [37]. 

These data may reflect a mechanism whereby increased RT doses damage the amygdala and its networks supporting an ERT. RT doses to the amygdala are associated with volume loss in the amygdala. Future work may investigate whether change on ERT scores correlates with volume loss or other imaging-based biomarkers for RT damage in the amygdala and associated structures.

This study has several limitations. First, an ERT cannot reflect all social cognition; however, it is an essential building block for more complex social cognition tasks; therefore, it may reflect dysfunction in social cognition that could impact patients’ quality of life. Additionally, this is a small patient sample size, which may limit our ability to identify factors that could influence the impact of RT doses on the amygdala, such as patient age. In order to recruit sufficient patients at a single institution, this cohort included multiple tumor types and tumor locations; however, we limited our cohort to low-grade and benign tumors, which will have less of a baseline impact on cognition due to slow growth and a less infiltrative pathology, minimizing this issue. The variety of tumor types also makes our results more generalizable. In addition, the longitudinal study design allows us to adjust for baseline scores, which is highly important in this type of study. A small percentage of these patients also had chemotherapy exposure, which could be a source of confounding as well. Finally, this study examined the impact of the mean RT dose on intracranial structures, which does not capture other RT planning parameters, such as radiation doses to a specific volume of a given structure. 

This result suggests multiple directions for future studies. Given that previous studies have shown that RT dose to the amygdala correlates with scores on emotional well-being [37], future studies may evaluate how social and emotional well-being correlates with changes in ERT outcomes. Future studies should incorporate normal tissue complication probability models, allowing a more detailed examination of how RT doses are associated with toxicities. Finally, the correlation of this outcome with MRI biomarkers of amygdala and orbitofrontal cortex RT damage, such as atrophy, will be important in confirming these results.

## 5. Conclusions

In conclusion, patients with brain tumors frequently have cognitive dysfunction, which can significantly impact their quality of life. While some of this is related to the tumor, cancer treatments such as RT may also contribute to cognitive dysfunction. Social cognition is an understudied but crucial cognitive domain that is impacted in patients with brain tumors. Our study suggests that an off-target RT dose to the amygdala–orbitofrontal network may further contribute to social cognition dysfunction in patients with brain tumors who are treated with RT.

## Figures and Tables

**Figure 1 cancers-15-05544-f001:**
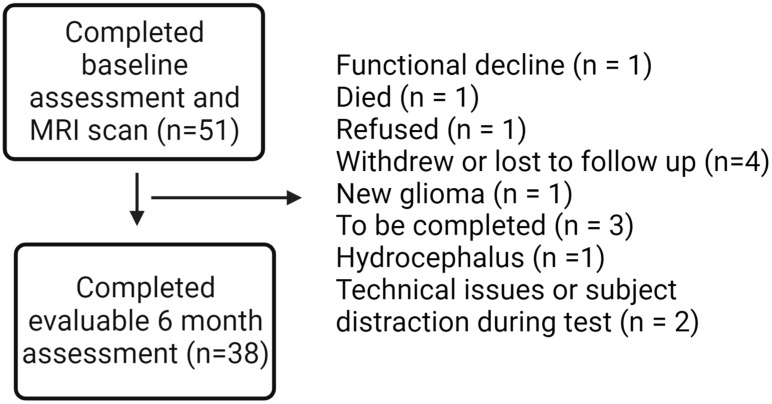
Consort diagram.

**Figure 2 cancers-15-05544-f002:**
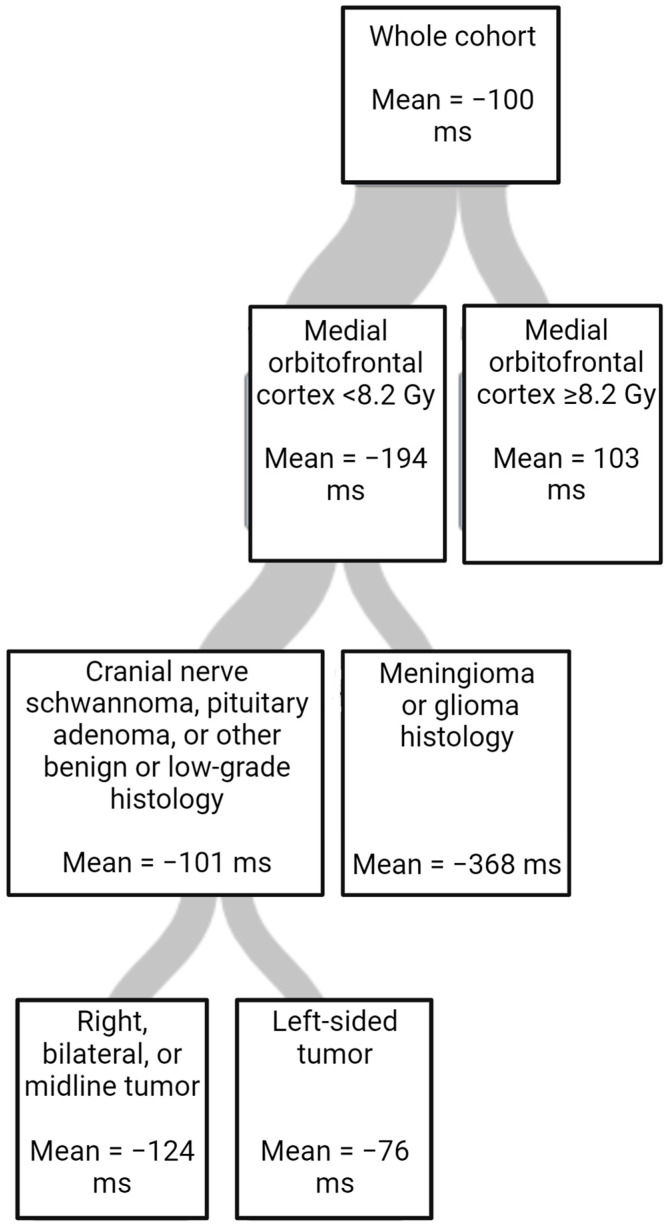
Regression tree for change in ERTOMDCRT scores in milliseconds (ms) fitted using random forest (RSS criterion, cost-complexity pruning, 10-fold cross-validation, minimum left size = 8).

**Table 1 cancers-15-05544-t001:** Distribution of baseline patient characteristics and association with median-correct reaction time (ERTOMDCRT) at 6 months.

Variable	Baseline Cohort n = 51 N (%) or Median (IQR *)	Subjects with Evaluable 6-Month Data n = 38 N (%) or Median (IQR *)	Median 6-Month ERTOMDCRT (IQR *) or Correlation Coefficient	*p*-Value
Whole Cohort	51 (100%)	38 (100%)	1381.3 (498.5)	.
Sex				0.73 ^
Male	20 (39%)	14 (37%)	1381.3 (529.5)	
Female	31 (61%)	24 (63%)	1357.3 (491.3)	
Handedness				0.44 ^
Right	46 (90%)	34 (89%)	1406.3 (490.0)	
Left or ambidextrous	5 (10%)	4 (11%)	1133.8 (375.8)	
Tumor histology				0.02 ^
Meningioma	20 (39%)	15 (39%)	1421.0 (344.0)	
Schwannoma	16 (31%)	13 (34%)	1095.5 (431.0)	
Glioma	5 (10%)	3 (8%)	1619.5 (1290.0)	
Pituitary adenoma	5 (10%)	3 (8%)	1570.0 (899.5)	
Other benign or low-grade tumor	5 (10%)	4 (11%)	1797.8 (757.5)	
Tumor grade				0.45 ^
Grade 1 or presumed grade 1 via imaging	39 (73%)	29 (76%)	1297.0 (501.0)	
Grade 2 or n/a	12 (22%)	9 (24%)	1533.5 (340.5)	
Tumor laterality				0.07 ^
Right	18 (35%)	12 (32%)	1229.8 (442.3)	
Left	22 (43%)	16 (42%)	1285.3 (528.0)	
Bilateral or midline	11 (22%)	10 (26%)	1558.5 (545.5)	
Race				0.42 ^
White	43 (84%)	32 (84%)	1267.5 (557.5)	
Black or African-American	6 (12%)	5 (13%)	1480.0 (73.0)	
Asian or Pacific Islander	2 (4%)	1 (3%)	1203	
Received chemotherapy				0.45 ^
Yes	4 (8%)	3 (8%)	1619.5 (1290.0)	
No	47 (92%)	35 (92%)	1367.5 (483.5)	
Surgical resection				0.08 ^
Yes	19 (37.2%)	13 (34.2%)	1545.0 (398.0)	
No	32 (62.8%)	25 (65.8%)	1238.0 (534.0)	
Time from surgery to RT start (months)	15.7 (37.5)	14.9 (36.1)	0.29	0.35 **
Total prescribed RT dose (Gy)	21 (36)	23 (36)	0.37	0.02 **
Total prescribed RT dose category				0.08 ^
>20 Gy	29 (57%)	23 (61%)	1472.0 (416.5)	
≤20 Gy	22 (43%)	15 (39%)	1179.0 (543.5)	
Number of RT fractions (all)	3 (26)	3 (26)	0.33	0.05 **
Type of fractionation				0.20 ^
Conventional fractionation (>5 fractions)	22 (43%)	17 (45%)	1533.5 (348.5)	
Hypofractionation (≤5 fractions)	29 (57%)	21 (55%)	1233.0 (434.0)	
Age (years)	58 (17)	58 (18)	0.49	0.0016 **
Age category				0.01 ^
≥60 years	22 (43%)	16 (42%)	1546.0 (605.8)	
<60 years	29 (57%)	22 (58%)	1191.0 (552.0)	
PHQ2 (all)	1 (2)	1 (2)	0.04	0.85 **
PHQ2 score category				0.39 ^
≥2	16 (31%)	13 (34%)	1238.0 (577.0)	
<2	35 (69%)	25 (66%)	1395.0 (490.0)	
Baseline ERTOMDCRT			0.62	<0.0001 **

^ Wilcoxon rank sum test; * interquartile range; ** Spearman correlation.

**Table 2 cancers-15-05544-t002:** Changes in ERTOMDCRT, ERTHH, and ERTOMDRT and association with doses to the amygdala, lateral orbitofrontal cortex, and medial orbitofrontal cortex after adjusting for baseline scores.

	ERTOMDCRT		ERTTH		ERTOMDRT	
	Beta	*p*-Value	Beta	*p*-Value	Beta	*p*-Value
Amygdala						
Right	5.94	0.25	−0.06	0.32	13.40	0.08
Left	7.34	0.07	−0.07	0.19	13.12	0.03
Highest mean RT dose	7.20	0.05	−0.05	0.27	12.56	0.02
Lateral orbitofrontal cortex						
Right	5.55	0.17	−0.07	0.14	14.30	0.02
Left	6.65	0.10	−0.09	0.07	5.77	0.02
Highest mean RT dose	6.16	0.08	−0.06	0.14	12.96	0.01
Medial orbitofrontal cortex						
Right	7.73	0.09	−0.10	0.08	18.58	0.01
Left	7.80	0.08	−0.10	0.05	17.15	0.01
Highest mean RT dose	7.98	0.07	−0.10	0.07	17.30	0.01

**Table 3 cancers-15-05544-t003:** Difference in change scores for ERTOMDCRT, ERTOMDRT, and ERTHH at the cut points defined on the regression trees for the amygdala, lateral orbitofrontal cortex, and medial orbitofrontal cortex.

	Change in Score from Baseline to 6 Months (ms or Total Count, Interquartile Range)	*p*-Value
ERTOMDCRT		
Whole cohort	−60.0 (458.0)	n/a
Highest mean RT dose to amygdala ≥ 6.5 Gy **		0.03 ^^
Yes (n = 13)	−6.5 (356.0)	
No (n = 25)	−132.0 (509.5)	
Highest mean RT dose to medial orbitofrontal cortex ≥ 8.2 Gy **		0.02 ^^
Yes (n = 12)	94.3 (438.0)	
No (n = 26)	−132.5 (509.5)	
Highest mean RT lateral orbitofrontal cortex ≥ 4.9 Gy **		0.09 ^^
Yes (n = 16)	−54.8 (452.8)	
No (n = 22)	−108.5 (617.5)	
ERTOMDRT		
Whole cohort	−60.0 (458.0)	n/a
Highest mean RT dose to amygdala ≥ 0.6 Gy **		0.01 ^
Yes (n = 26)	28.0 (436.0)	
No (n = 12)	−369.3 (980.8)	
Highest mean RT dose to medial orbitofrontal cortex ≥ 1.1 Gy **		0.02 ^
Yes (n = 17)	40.5 (436.0)	
No (n = 21)	−161.5 (411.5)	
Highest mean RT dose to lateral orbitofrontal cortex ≥ 1.1 Gy **		0.04 ^
Yes (n = 18)	31.5 (441.5)	
No (n = 20)	−148.3 (656.5)	
ERTHH		
Whole cohort	2.0 (5.0)	n/a
Highest mean RT dose to amygdala ≥28 Gy **		0.92 ^^
Yes (n = 4)	5.0 (13.0)	
No (n = 34)	2.0 (5.0)	
Highest mean RT dose to medial orbitofrontal cortex ≥ 17.4 Gy **		0.05 ^^
Yes (n = 4)	−2.0 (10.5)	
No (n = 34)	2.5 (5.0)	
Highest mean RT dose to lateral orbitofrontal cortex ≥ 29.7 Gy **		0.05 ^^
Yes (n = 4)	−2.0 (10.5)	
No (n = 34)	2.5 (5.0)	

^ Wilcoxon; ^^ *t*-test; ** mean RT dose after conversion to EQD2.

**Table 4 cancers-15-05544-t004:** Results from multivariate linear regression predicting 6-month ERTOMDCRT score.

Parameter	Estimate	Standard Error	*p* > |t|
Baseline ERTOMDCRT score	0.48	0.11	0.0002
Highest mean RT dose to amygdala (Gy)	7.84	3.46	0.03
Age	6.56	4.59	0.16
Histology	69.12	45.71	0.14
Tumor laterality	−26.86	78.42	0.73
PHQ2 score (depression)	−15.72	36.82	0.67
Sex	−35.99	100.88	0.72

**Table 5 cancers-15-05544-t005:** Variable importance ranked using a random forest regression tree for change from baseline to 6 months in ERTOMDCRT scores.

	Training	
Variable	Relative	Importance
Highest mean RT dose to medial orbitofrontal cortex	1.00	849.4
Primary tumor histology	0.76	646.8
Tumor laterality (right, left, or midline)	0.12	98.8

## Data Availability

The data presented in this study are available upon request from the corresponding author. The data are not publicly available due to concerns about participant privacy.

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
