# Peer review of "Association of Radiation Dose to the Amygdala–Orbitofrontal Network with Emotion Recognition Task Performance in Patients with Low-Grade and Benign Brain Tumors"

_cancers, 2023, doi:10.3390/cancers15235544_

Round 1

Reviewer 1 Report

Comments and Suggestions for Authors

Hardy et al. provide an excellent initial study evaluating this emerging area of interest in CNS radiation oncology. I appreciate the detailed methodology provided in the manuscript, which will be drawn upon for future analyses on this and related topics. 

The topic explores a newly emerging area of interest in the field regarding the neurocognitive impact of integral dose received by specific intracranial structures and provides initial data to consider larger prospective, potentially randomized, studies.

For the purposes of this initial analysis, no further improvements. Future studies can look into actively sparing these structures, as well and the hippocampi, and if there is a benefit to only sparing contralateral structures. Other considerations are separating benign and malignant histologies. Schwannomas are almost universally localized to the 8th cranial nerve, and should be considered separately.

The conclusions is consistent with the evidence and arguments presented and they address the main question
"Is radiotherapy dose to the amygdala-orbitofrontal network associated with performance on a social cognition tasks".

The references are appropriate.

Reviewer 2 Report

Comments and Suggestions for Authors

The authors prospectively enrolled low-grade and benign brain tumors patients who received cranial RT. They assessed longitudinal changes on an emotion recognition task (ERT) and association with RT dose to the amygdala-orbitofrontal network. As results, RT dose to the amygdala-orbitofrontal network was significantly associated with longer response times on the ERT, and the medial orbitofrontal cortex RT dose was the most important variable on regression trees predicting change in the ERTOMDCRT.

This is an interesting paper regarding RT dose to the amygdala-orbitofrontal network and changes of cognitive function. However, it has some major issues that should be resolved.

1.      Half of the data include hypofractionated RT, but the authors only analyzed the (highest) total dose in the amygdala-orbital frontal network; various RT parameters, including mean dose per fraction, and total dose (2 Gy equivalent) as well as total dose, should be analyzed.

2.      In the Methods section, the authors note that "the amygdala, lateral orbitofrontal cortex, and medial orbitofrontal cortex are paired structures. Thus, for an individual patient, we were interested in the impact of higher RT dose to at least one of the pair, and we recorded the highest mean dose to the pair as a separate variable called Highest Mean RT Dose. "  
  What does "Mean" mean in this terminology? This could be misleading to the reader.

3.      In the Methods section, the authors note that clinical variables were recorded, including age, education level, comorbidities including diabetes, hypertension, autoimmune disease, tumor hemisphere, tumor site, tumor pathology, prior surgeries, employment status, smoking status, alcohol use, sex/gender, hypopituitarism, menopausal status, steroid use, use of medications that can affect cognition and mood, and exposure to chemotherapy. Some data, however, are not listed in the Results section. At the very least, a history of prior surgery and the time from resection to the start of RT should be indicated.

4.      Please provide some representative figures of a patient with and without cognitive impairment at 6 months post-RT.

Round 2

Reviewer 2 Report

Comments and Suggestions for Authors

The manuscript has been adequately revised.